# Soil Nematodes Regulate Ecosystem Multifunctionality Under Different *Zokor Mounds* in Qinghai–Tibet Alpine Grasslands

**DOI:** 10.3390/biology14091200

**Published:** 2025-09-05

**Authors:** Xiaodong Zhang, Lili Nian, Liangliang Li, Xuelu Liu, Qi Wang

**Affiliations:** 1Pratcultural College, Gansu Agricultural University, Lanzhou 730070, China; m18409492603@163.com (X.Z.); 18394797671@163.com (L.L.); 2Institute of Soil, Fertilizer and Water-Saving Agriculture, Gansu Academy of Agricultural Sciences, Lanzhou 730070, China; 18893814845@163.com; 3College of Resources and Environmental Sciences, Gansu Agricultural University, Lanzhou 730070, China

**Keywords:** ecosystem multifunctionality, metabolic footprint, Qinghai–Tibet Plateau, soil nematodes, *zokor mound*

## Abstract

**Simple Summary:**

The Qinghai–Tibet Plateau is a fragile high-altitude ecosystem where alpine grasslands are threatened by climate change and human disturbance. Plateau *zokors* create soil mounds that alter habitat conditions, but their effects on soil life and ecosystem functioning are poorly understood. We studied four mound types: undisturbed grassland, Potentilla anserina mounds, Leontopodium mounds, and bare new mounds. Soil nematodes were extracted and analyzed for abundance, diversity, functional indices, and metabolic footprints. Compared with undisturbed grassland, Potentilla mounds increased nematode abundance by up to 39%, while bare mounds raised dominance and bacterial pathway indicators but reduced diversity by up to 29%. Leontopodium mounds lowered nematode maturity and raised plant parasite levels. Soil nutrient cycling and multifunctionality were highest in undisturbed grassland and lowest in Leontopodium mounds. Moisture enhanced nitrogen and phosphorus cycling, while higher pH reduced carbon cycling. Overall, vegetation-covered *zokor mounds* influenced nutrient cycles and ecosystem multifunctionality by altering nematode communities. These findings improve our understanding of soil biodiversity–function relationships and provide practical guidance for restoring degraded alpine grasslands.

**Abstract:**

The Qinghai–Tibet Plateau’s alpine grasslands are ecologically vulnerable. Plateau *zokors* build mounds that modify soil and vegetation, influencing soil biota. This study examined how different vegetation on *zokor mounds* affects soil nematodes and ecosystem function. We compared undisturbed grassland (CK), Potentilla anserina (PM) and Leontopodium (LM) mounds, and new bare mounds (NM). Soil nematode communities were analyzed to assess functional indices and metabolic footprints. Compared with CK, PM increased total nematode abundance by 37.74%, r-strategists by 36.54%, and K-strategists by 39.37%. NM increased dominance (λ) by 22.20%, channel ratio (NCR) by 8.89%, and the Wasilewska index (WI) by 1.24 times, but reduced Shannon diversity by 8.49%, trophic diversity (TD) by 22.84%, and species richness (SR) by 29.40%. LM decreased the maturity index (MI) of free-living nematodes by 7.19% and increased the plant parasite index (PPI) by 10.01%. PM exhibited the highest metabolic footprints for bacterivores, fungivores, omnivores/predators, and total nematodes. Soil carbon (EF-C), nitrogen (EF-N), phosphorus (EF-P) cycling functions, and overall ecosystem multifunctionality (EMF) were highest in CK and lowest in LM. Soil moisture had positive effects on EF-N, EF-P, and EMF, whereas pH had a negative effect on EF-C. These findings demonstrate that vegetation-covered *zokor mounds* influence nutrient cycling and ecosystem multifunctionality through changes in nematode community characteristics, providing new insights into soil biodiversity–function relationships and informing grassland restoration strategies in high-altitude ecosystems.

## 1. Introduction

Due to its unique geographical location and complex climatic conditions, the Qinghai–Tibet Plateau is widely recognized as the “Third Pole of the World.” It serves as a critical water conservation area and climate regulation zone in East Asia and globally, while also functioning as China’s largest alpine pastoral region and an essential ecological barrier [1,2,3]. The region encompasses a variety of ecosystem types, with alpine grasslands constituting the predominant type, covering approximately 50% of the total area [4,5]. In recent decades, a combination of factors, such as increasing rodent populations, excessive grazing pressure, and global climate change, has led to widespread degradation of alpine grasslands on the plateau [6,7,8]. It is estimated that 50.4% of natural grasslands in the region have experienced degradation, with 16.5% classified as severely degraded [9,10]. *Zokor mounds* are formed by the burrowing of plateau *zokors* (*Eospalax baileyi*). They are loose, nutrient-rich soil patches, 40–150 cm in diameter and 20–60 cm in height. By altering soil structure and hydrothermal conditions, these mounds become biodiversity hotspots that influence plant communities and soil nutrient functions [11,12,13]. Under low to moderate grazing pressure, *zokor* disturbance can enhance plant species diversity, accelerate nutrient cycling, and increase habitat heterogeneity, thereby earning them the designation of “ecosystem engineers” in grassland systems [14,15]. However, under overgrazing, zokor activity can severely damage native vegetation. It also triggers the spread of toxic and unpalatable weeds. This process transforms high-quality pastures into low-productivity “black-soil patches” and becomes a major driver of alpine grassland degradation [16].

Soil nematodes represent one of the most vital and dynamic components of grassland soils [17]. They play essential roles in regulating plant growth, mediating soil ecological processes, and sustaining the functionality of grassland ecosystems [18,19]. Previous studies have mainly examined the links between soil microbial communities and grassland ecosystem functions. However, new evidence shows that nematode communities also play an important role. In particular, higher-trophic-level nematodes strongly influence multiple ecosystem functions [20]. As integral elements of the soil food web, nematodes are categorized into several trophic groups based on their feeding behaviors, including bacterivores, fungivores, herbivores, and omnivorous predatory types [21]. They take part in almost all soil ecological processes. Their community composition directly shapes soil food web structure, ecosystem functions, and overall soil health [22].

Soil multifunctionality is fundamental to sustaining ecosystem services. On the Qinghai–Tibet Plateau, it is mainly driven by soil nutrients. Key functions related to carbon, nitrogen, and phosphorus strongly influence vegetation growth and guide ecological restoration [20]. Carbon and nitrogen are major soil nutrients that support plant growth and provide energy for microbial activity. They also regulate soil moisture and temperature. In addition, both elements help maintain the stability of soil physical structure [23]. Therefore, evaluating nutrient-based soil multifunctionality is fundamental for understanding ecological restoration in artificial grasslands across the Qinghai–Tibet Plateau. Soil nematodes are important drivers of soil functional expression. Understanding the relationship between soil nematodes and soil multifunctionality is essential for assessing the impact of *zokor mounds* on soil ecological processes. Recent studies increasingly highlight the link between soil biodiversity and ecosystem multifunctionality. However, most have focused only on single-trophic-level microorganisms. Far less attention has been given to nematode communities, which are multi-trophic groups with complex network structures [24]. Emerging evidence suggests that variations in nematode abundance significantly influence grassland ecosystem multifunctionality [25]. Furthermore, nematodes contribute to soil organic matter decomposition, nutrient cycling, disease suppression, and potentially to increased crop yields [26]. Therefore, the abundance, diversity, and network complexity of nematode communities are crucial biological factors underpinning improvements in the multifunctionality of agroecosystems [27].

This study examined the impact of zokor activity on grassland ecosystems in the Qinghai–Tibet Plateau and investigated how soil nutrient dynamics influence nematode diversity and ecosystem multifunctionality. This study tested two hypotheses: (1) The density, diversity, community composition, and metabolic footprint of soil nematodes in grasslands are influenced by *zokor mounds* with different vegetation characteristics. (2) Soil nutrient alterations caused by *zokor* disturbance indirectly affect nematode diversity, which subsequently influences ecosystem multifunctionality. To evaluate these hypotheses, nematode community diversity and structure in different *zokor mounds* were assessed using morphological identification techniques. Inter-genus relationships were examined through molecular ecological network analysis, and soil ecosystem multifunctionality and its primary environmental drivers were quantified using the averaging method. These findings provide a theoretical foundation for understanding soil nematode ecology and promoting sustainable grassland management on the Qinghai–Tibet Plateau.

## 2. Study Areas and Methods

### 2.1. Study Area

The experimental site is located in Wanmao Town, Zhuoni County, Gannan Tibetan Autonomous Prefecture, Gansu Province, China (103°03′32.68″ E, 34°51′42.87″ N), at an elevation of 3040 m. The region exhibits a typical plateau continental climate, characterized by a mean annual temperature of 2.3 °C and an average frost-free period of approximately 105 days. The mean annual precipitation is 640 mm, with around 40 precipitation days per year. In extreme years, annual precipitation ranges from a minimum of 400 mm to a maximum of 800 mm. Precipitation is highly seasonal, predominantly occurring between May and August, with July receiving the highest rainfall. The vegetation type is mainly alpine meadow. The dominant species is *Kobresia pygmaea* of the Cyperaceae family. The main associated species include *Elymus nutans* and *Poa pratensis* of the Poaceae family, *Anemone rivularis* of the Ranunculaceae family, *Potentilla anserina* and *Potentilla fragarioides* of the Rosaceae family, and *Saussurea japonica* of the Asteraceae family.

### 2.2. Sample Collection

In August 2022, we conducted a random selection of alpine meadows with relatively flat and consistent terrain that experienced fencing in warm seasons and grazing in cold seasons as the survey area (Figure 1). Within this survey area, we identified three types of *zokor mounds* based on the dominant plant groups: *Potentilla anserina zokor mounds* (PM), *Leontopodium leontopodioides zokor mounds* (LM), and bare new *zokor mounds* (NM). We selected four *zokor mounds* from each type to form four repetitions. Additionally, following the principle of proximity, we selected a control sample circle (CK) in undisturbed grassland within the same area where *zokor* activity was absent. All selected *zokor mound* colonies were of similar size to CK (approximately 50 cm in diameter).

In August 2022, five soil cores (0–20 cm in depth, 2 cm in diameter) were randomly collected from each selected zokor mound and control plot. The five cores from each plot were homogenized to form a composite sample. A total of 16 composite soil samples were obtained (4 treatments with 4 replicates each). After collection, the soils were homogenized and passed through a 2 mm sieve, with visible plant debris carefully removed. The processed soils were then split into three portions. One portion was transferred into sterile centrifuge tubes, immediately packed in an ice-cooled container, and transported to the laboratory, where it was preserved at −80 °C for subsequent DNA extraction. Another portion was sealed in plastic bags, transported in a cooler, and stored at 4 °C to determine the soil water content and microbial biomass carbon and nitrogen. The remaining portion was air-dried under ventilation and reserved for the measurement of soil pH, electrical conductivity, organic carbon, total nitrogen, total phosphorus, available phosphorus, and available potassium.

### 2.3. Soil Physicochemical Analysis

Soil organic carbon (SOC) was determined using the dichromate oxidation method [28]. Microbial biomass carbon (MBC) [29] and nitrogen (MBN) [30] were obtained using the chloroform fumigation–extraction method, with concentrations analyzed on a TOC/TN analyzer (Multi C/N 2100S, Analytik Jena, Jena, Germany). Total nitrogen (TN) was measured by the Kjeldahl method [31], and total phosphorus (TP) by H_2_SO_4_–HClO_4_ digestion [32]. Available phosphorus (AP) was extracted with 0.5 M NaHCO_3_ and determined colorimetrically [33], while available potassium (AK) was extracted with 1 M NH_4_OAc and analyzed by flame photometry [34].

### 2.4. Identification of Soil Nematode Communities

Soil nematodes were isolated using the shallow dish technique. For each extraction, 100 g of soil was spread on filter paper in a shallow dish, and tap water was carefully added along the dish wall until a thin film covered the soil surface. The samples were kept at room temperature for 48 h, after which nematodes were recovered using a 500-mesh sieve (25 μm). Extracted nematodes were then killed in a 60 °C water bath, preserved in 4% formalin, and stored in labeled vials for later identification [35]. Identification was based on morphological traits, particularly head features and feeding habits, classifying nematodes into four trophic groups: bacterivores, fungivores, plant parasites, and omnivores/predators. According to life-history strategies, they were further divided into colonizers (r-strategists) and persisters (K-strategists). Each taxon was assigned a colonizer–persister (c-p) score from 1 (fast-growing r-strategists) to 5 (slow-growing K-strategists), representing their ecological strategy and disturbance tolerance [36].

### 2.5. Calculation of Nematode Ecological Function Index and Metabolic Footprint

The nematode diversity was estimated by the Shannon–Wiener diversity index (H’), Simpson dominance index (λ), trophic diversity index (TD), Margalef index (SR), free-living nematode maturity index (MI), plant-parasitic nematode maturity index (PPI), nematode channel ratio (NCR), and Wasilewska index (WI), as follows [37]:$$H^{\prime }=\;-\sum_{i=1}^{s}pi*lnpi$$$$\Lambda =\sum {pi}^{2}$$$$\mathrm{TD}=1/\sum {pi^{\prime }}^{2}$$SR=(S – 1)/lnN$$\mathrm{MI}=\sum v_{i}f_{i}$$$$\mathrm{PPI}=\sum v_{i}f_{i}^{\prime }$$$$\mathrm{NCR}=\frac{BF}{BF+FF}\;$$$$\mathrm{WI}=\frac{BF+FF}{pp}\;$$

Here, pi denotes the relative abundance of taxon i within a sample, and S represents the total number of nematode genera in the community. pi′ indicates the proportion of the abundance of taxon i within its trophic group relative to total nematode abundance. $v_{i}$ is the colonizer–persister (c-p) value assigned to taxon i. $f_{i}$ corresponds to the proportion of free-living nematodes in taxon i, while $f_{i}^{\prime }$ refers to the proportion of plant-parasitic nematodes. BF, FF, and PP represent the abundances of bacterivores, fungivores, and plant parasites, respectively.

The nematode metabolic footprint was calculated using the biomass (fresh weight, *W*) of various nematode groups listed in the Nematode-Plant Expert Information System (http://nemaplex.ucdavis.edu, (accessed on 2 May 2025)).$$\mathrm{NMF}=\Sigma (N_{t}\;\times \;(0.1\;\times \;(W_{t}\;÷{\;m}_{t})+0.273(W_{t}^{0.75})))$$

Here, $N_{t}$ denotes the abundance of the *t*-type nematode population; $m_{t}$ is its corresponding c-p value; and $W_{t}$ represents the biomass of the *t*-type nematode population [37].

The functional metabolic footprint (FMF) is expressed as the area occupied by the enrichment footprint and the structural footprint, with the formula (Fs × Fe)/2, which can measure the ability of a food web to regulate and maintain its metabolic balance [37]. That is, the larger the functional footprint, the greater the contribution of the nematode to regulating the food web, and the stronger the ability of the predator and the prey to maintain their metabolic balance [37,38].

### 2.6. Calculation of Soil Ecosystem Multifunctionality

In this study, fifteen ecosystem function (EF) indicators were assessed and classified into four categories: (1) carbon-related functions (EF-C) comprise soil organic carbon, microbial carbon, easily oxidizable organic carbon, soluble organic carbon, and particulate organic carbon; (2) nitrogen-related functions (EF-N) include soil total nitrogen, soil microbial nitrogen, acid-hydrolyzable total nitrogen, acid-hydrolyzable ammonium nitrogen, acid-hydrolyzable amino acid nitrogen, amino sugar nitrogen, and acid-hydrolyzable unknown nitrogen; (3) phosphorus-related functions (EF-P) include soil total phosphorus and available phosphorus; (4) ecosystem multifunctionality (EMF) is an integrative index based on all 15 ecosystem function indicators. These indicators were chosen for their essential roles in regulating and sustaining major ecological processes in grassland ecosystems, and they are commonly used in studies of ecosystem functioning and multifunctionality [39].

The ecosystem multifunctionality index (*EMF*) was derived using the arithmetic mean method [40]. Firstly, the above 15 ecological function indicators were standardized.$$f_{ij}\;=\frac{x_{ij}-{min}_{ij}}{{max}_{ij}-{min}_{ij}}$$

Here, $f_{ij}$ represents the standardized value of the j-th ecosystem function at sample site i, $x_{ij}$ is the observed value of that function; ${min}_{ij}$ and ${max}_{ij}$ denote the minimum and maximum values of the *jjj*-th function across all sites under the same factor, respectively.

The ecosystem function index (*EF*) was then calculated using the single-function approach:$${EF}_{ij}=\sum_{j}^{n}\frac{f_{ij}}{n}$$

The ecosystem multifunctionality index (*EMF*) was computed using the mean method:$${EMF}_{i}=\frac{1}{N}\sum_{1}^{N}f_{ij}$$
where ${EF}_{ij}$ denotes the functional index of the j-th function in plot i; *n* is the number of ecosystem indicators included within that function; *EMF* represents the ecosystem multifunctionality index of plot i, obtained as the standardized mean of all indicators; and *N* is the total number of ecosystem functions considered in plot i.

### 2.7. Data Processing

Differences in nematode ecological function indices, abundance, trophic groups, and multifunctionality were tested with one-way ANOVA in SPSS 26.0, followed by multiple comparisons using the LSD method at *p* = 0.05. Principal component analysis (PCA) was performed in R (version 1.6.0.) with the “stats” package to qualitatively assess similarities and differences in nematode community composition and to identify major components driving variation among groups. To further assess whether observed differences in species composition were statistically significant, Analysis of Similarity (ANOSIM) was performed using the “vegan” package in R. Structural Equation Modeling (SEM) was applied via the “piecewiseSEM” package in R to elucidate the relationships among nematode trophic groups and soil physicochemical properties, providing a mechanistic understanding of their interactions. Bar plots depicting the metabolic footprint of different nematode trophic groups were generated using Origin 2021, offering insight into the relative carbon utilization intensity among groups. Final graphical modifications were completed using Adobe Illustrator (version 2024) to ensure high-quality visual presentation.

## 3. Results and Analysis

### 3.1. Soil Nematode Community Composition

A total of 29,280 nematodes were identified in 16 soil samples, with an average density of 1830/100 g dry soil (Figure 2a). The distribution of dominant genera (relative abundance > 10%), common genera (1–10%), and rare genera (<1%) of soil nematodes was significantly different among the different treatments. The dominant genus in the control (CK) was the fungivorous nematode *Filenchus* (10.75%); in the NM treatment, the bacterivorous nematode *Isolaimium* (10.5%) and the fungivorous nematode *Filenchus* (10.5%) were both dominant genera; in the PM treatment, *Isolaimium* was the only dominant genus (10.75%); no dominant genus appeared in the LM treatment. The relative abundance differences of various trophic groups in the nematode community were analyzed by radar charts (Figure 2b). The results show that the relative abundance of bacterivorous nematodes and omnivorous/predatory nematodes in the NM treatment were significantly higher than that in other treatments; the relative abundance of plant-parasitic nematodes was the highest in the LM treatment; and the relative abundance of fungivorous nematodes was higher in the PM treatment. The Venn diagram analysis of the overlap and differences in nematode genera among different treatments showed that there were 19 nematode genera in the four treatments of CK, NM, PM, and LM, and no unique genera were detected in each treatment (Figure 2c).

The abundance of soil nematodes was significantly different among different treatments (Figure 2d). Compared with the CK treatment, the PM treatment increased total nematode abundance, cp1–2 group abundance, and cp3–5 group abundance by 37.74%, 36.54%, and 39.37%, respectively. The results of principal component analysis (PCA) show that there were significant differences in the nutritional groups, colonizer–persister (c-p) groups, and the total nematode community among the different zokor mound treatments (*p* < 0.05), indicating that *zokor mound* disturbance had a significant effect on the structure of the soil nematode community (Figure 3).

The diversity index and ecological function index of soil nematode community were significantly different among the treatments (*p* < 0.05) (Figure 4a). Compared with the CK treatment, the NM treatment increased the nematode dominance index (λ) and nematode channel ratio (NCR) by 22.20% and 8.89%, respectively, and increased the Wasilewska index (WI) by 1.24 times. In contrast, the Shannon diversity index, trophic diversity index (TD), and species richness (SR) of nematodes decreased by 8.49%, 22.84%, and 29.40%, respectively. Compared with CK, the LM treatment significantly reduced the maturity index (MI) of free-living nematodes by 7.19% and significantly increased the plant-parasite index (PPI) by 10.01%.

*Zokor mounds* under different vegetation types had a significant effect on the characteristics of soil nematode co-occurrence networks (Figure 4b). The results of co-occurrence network analysis show that the number of nodes, number of connections, average degree, and modularity index in the NM treatment were the lowest, indicating that the complexity of its nematode community co-occurrence network was low, the interactions between species were relatively simple, and the overall network structure was less stable. In contrast, the number of nodes, number of connections, and average degree in the PM treatment were the highest, indicating that the nematode community co-occurrence network structure was more complex under this treatment, and the potential interactions between species were closer. In addition, the number of positively correlated edges in the LM treatment was significantly lower than that in other treatments, indicating that the nematode community in this treatment was dominated by mutual competition and weak cooperative interactions.

### 3.2. Metabolic Footprint and Flora Analysis of Soil Nematodes

The metabolic footprints of bacterivores, fungivores, omnivores/predators, and total nematodes were significantly greater in the PM treatment than in the other treatments (*p* < 0.05), suggesting enhanced carbon utilization and higher energy flow efficiency of the ecosystem under this condition. Although plant-parasitic nematodes showed no significant differences across treatments, their footprint values in the PM treatment were still higher than those in the others, indicating elevated activity as well (Figure 5).

The flora structure analysis revealed that CK sample points were primarily located in quadrant B, NM and PM samples were distributed across quadrants B and C, and LM samples were mainly clustered in quadrant C (Figure 6). These results show that different treatments have significant effects on the structure and functional construction of soil nematode communities, and that the nutrient enrichment effect and soil food web structuring level of *zokor mound* soil are generally low, with the LM treatment being the lowest. From the perspective of functional footprint graphical features (such as diamond area), the functional footprint of nematodes in the PM treatment was significantly larger than that in other treatments, further indicating that its ecological function was more complete. In terms of structural index, the structural index of the PM and NM treatments was slightly higher than that of the CK treatment, showing better community stability and complexity. Meanwhile, the structural index of the LM treatment was significantly lower than that of CK, indicating that the nematode community structure was relatively simple and the food web stability was poor under this treatment.

### 3.3. Soil Ecosystem Multifunctionality

The single functions of the zokor mound grassland ecosystem under different vegetation types were evaluated, and the results are presented in (Figure 7). Significant differences were observed in EF-C, EF-N, EF-P, and EMF across treatments (*p* < 0.05). Specifically, CK showed the highest values for all four indices, whereas LM exhibited the lowest, suggesting that vegetation type on *zokor mounds* strongly influences soil ecological functions.

In order to further explore the impact path of different vegetation *zokor mounds* on ecosystem multifunctionality, a structural equation model was constructed (Figure 8d). The results show that soil water content and omnivorous/predatory nematode metabolic footprint had a significant positive effect on grassland EMF (*p* < 0.05). In the single-function path model of the ecosystem: The EF-C model showed that the soil pH and plant-parasitic nematode metabolic footprint had a significantly negative effect on carbon nutrient function (*p* < 0.05) (Figure 8a). The EF-N model showed that the soil water content, omnivorous/predatory nematode metabolic footprint, and fungivorous nematode metabolic footprint all had a significantly positive effect on nitrogen nutrient function (*p* < 0.05) (Figure 8b). In the EF-P model (Figure 8c), the soil water content, omnivorous/predatory nematode metabolic footprint, and plant-parasitic nematode metabolic footprint were all significantly positively correlated with phosphorus nutrient function (*p* < 0.05).

### 3.4. Relationship Between Soil Nematodes and Ecosystem Multifunctionality

The results of redundancy analysis (RDA) show that *Ditylenchus*, *Mesorhabditis*, *Cephalobus*, and *Basiria* were significantly positively correlated with soil water content; *Alaimus*, *Odontolaimus*, *Aphelenchus*, and *Acrobeles* were significantly positively correlated with soil pH; *Filenchus*, *Steinernema*, *Eucephalobus*, *Prodesmodora*, and *Aphelenchoides* were positively correlated with EF-C, EF-N, EF-P, and EMF (Figure 9a). The results of the Mantel analysis show that bacterivorous nematodes and plant-parasitic nematodes were significantly positively correlated with soil total phosphorus content (*p* < 0.05), while environmental factors had no significant effect on omnivorous/predatory nematodes or fungivorous nematodes, indicating that these two types of nematode communities are regulated by multiple factors (Figure 9b).

## 4. Discussion

### 4.1. Effects of Zokor Mounds with Different Vegetation on the Composition of Soil Nematode Communities

The results of this study show that different types of *zokor mounds* significantly affected the abundance, diversity, richness, and community structure of soil nematodes. This phenomenon was mainly attributed to the regulatory effects of different *zokor mounds* on the physical and chemical properties of the soil. The nematode community showed obvious species selectivity for environmental factors, which was manifested as a preference or rejection of specific environmental conditions, resulting in significant differences in the composition of the nematode community. Further ANOSIM analysis verified the significant effects of *zokor mounds* on the various nutritional types of soil nematodes, cp1–2 groups, cp3–5 groups, and the overall nematode community, supporting the hypothesis of this study. The mechanism may include two aspects: on the one hand, the differences in the biomass and community structure of vegetation of different *zokor mound* types changed the quantity and quality of soil organic matter input, thereby affecting the food resource supply of nematodes [41]; on the other hand, the changes in soil structure and water and heat conditions caused by *zokor mounds* regulated the living environment of nematodes [42]. Specifically, PM type *zokor mounds* significantly increased the number of soil nematodes, while LM type significantly reduced the number of nematodes. This may be related to the fact that the formation of *zokor mounds* loosens the originally compacted soil, improving soil aeration and water permeability, thereby creating a more suitable habitat for nematodes [39]. In contrast, the LM type *zokor mounds* at the end of vegetation secondary succession gradually compacted the soil due to factors such as increased vegetation coverage and livestock trampling in winter, increasing soil bulk density and decreasing permeability. The appearance and growth of edelweiss has become an indicator of declining soil quality [43]. Moreover, the higher cp1–2 and cp3–5 abundances in PM-type mounds compared with other treatments suggest a distinctive positive ecological effect, providing ample nutrient supply while rapidly promoting soil ecosystems toward greater stability, complexity, and functional diversity.

The soil nematode community ecological index plays an important role in assessing the structure and function of the soil food web, monitoring changes in soil biodiversity under vegetation restoration, and judging the state of ecosystem recovery [44]. In this study, the Shannon index of nematodes in the NM treatment was significantly lower than that in other treatments. This may be due to the strong disturbance of the soil environment by zokor activities, such as the input of feces and plant residues, which led to the rapid reproduction and dominance of bacteria-feeding nematodes with strong tolerance, while inhibiting the survival of fungal-feeding and predatory nematodes [45]. Low MI values and high PPI values may indicate that the ecosystem has been strongly disturbed. Conversely, high MI values and low PPI values may indicate that the ecosystem is tending to be stable [46]. In this study, the MI in the LM treatment was significantly lower than that in other treatments, and the PPI was significantly higher than that in other treatments, indicating that soil ecosystem stability in the LM treatment was poor. When the WI value is higher than 1, it indicates that the soil is in good health, and the soil food web is dominated by bacteria-feeding nematodes and fungi-feeding nematodes. The more complex and diverse mineralization process indicates that the soil food web has higher complexity and stronger ecological functions. On the contrary, when the WI value is lower than 1, it indicates that the soil is in poor health, and the soil mineralization pathway is mainly manifested as energy transfer from plants directly to herbivorous nematodes [21]. In this study, the WI values of NM were significantly higher than those of other *mouse mounds*, indicating that, compared with other *mouse mounds*, the soil health of NM was good. This may be because the plateau zokor’s nest-building and mound-building behaviors played a positive role in improving the soil nutrient status in the early stage of nest construction, and the nutrient content of most soils increased. The nematode NCR can be used to indirectly evaluate the dominant group pathway of organic matter degradation in the soil. If its value is greater than 0.75, it means that the degradation pathway of soil organic matter is mainly bacterial pathway type; if it is less than 0.75, it means that the degradation pathway of soil organic matter is mainly the fungal pathway type [47]. In this study, the organic matter decomposition pathway of each treatment was dominated by the bacterial decomposition pathway.

### 4.2. Effects of Zokor Mounds with Different Vegetation on the Metabolic Footprint of Soil Nematodes

The metabolic footprint of nematodes, as a reflection of their carbon metabolism process, is not only a direct indicator of the response of nematode populations to resource changes, but also a direct display of the ecological functions and services contributed by nematode communities. It plays a key role in quantitatively evaluating the structural and functional dynamics of soil food webs [37]. The metabolic footprints of in the PM treatment were all higher than those in other treatments. This is because PM is in the middle stage of *zokor mound* succession. Compared with CK and LM, its soil permeability is higher, providing more favorable survival and reproduction conditions for micro-nematodes. At the same time, the establishment of vegetation also enhanced the metabolic activities of plant-parasitic nematodes. The nematode structure index is an important metric for characterizing the soil food web, as it reflects both food web connectivity and food chain length. Higher index values indicate stronger connectivity and longer chains within the soil ecosystem. In contrast, the enrichment index serves as an indicator of external nutrient inputs. The larger its value, the more external nutrient input the soil system receives [48]. Flora analysis showed that the structure index of PM and NM was slightly higher than that of CK, while the structure index of LM was significantly lower than that of CK. This indicates that the NM and PM food webs have the strongest regulatory capacity, the most complex and stable food web structures, while the LM soil food web has the weakest regulatory capacity and the food web is degraded.

### 4.3. Effects of Zokor Mounds with Different Vegetation on Soil Ecosystem Functions

*Zokor mounds* with different vegetation types influence the soil environment and microbial communities by altering soil physicochemical properties and microbial composition, which in turn affect the stability and multifunctionality of grassland ecosystems [49]. In this study, soil nutrient functions were significantly higher under the CK treatment, whereas LM showed the lowest values among all treatments. This may be because Leontopodium has strong nutrient competitiveness and potential allelopathic effects, which inhibits soil microbial activity and key enzyme functions. At the same time, the damage to soil structure caused by rodent digging activities further aggravates the loss and functional degradation of carbon, nitrogen, and phosphorus. The highest nutrient functions and ecosystem multifunctionality observed in the CK treatment may be due to the undisturbed soil structure [50]. In addition, soil nutrient functions are composed of soil physical and chemical indicators. The effects of *zokor mounds* with different vegetation types on soil physical and chemical indicators ultimately manifest as an impact on the ecosystem functions composed of the corresponding indicators. Soil water content has a significant positive effect on nitrogen nutrient function, phosphorus nutrient function, and ecosystem multifunctionality. This may be because water can significantly increase microbial activity, promote nitrogen mineralization and nitrogen fixation, and enhance phosphatase activity and improve phosphorus effectiveness [51]. In addition, water helps maintain soil aggregate structure, reduces nutrient leaching, and enhances ecosystem multifunctionality by promoting plant growth and microbial diversity [52]. Soil pH showed a significant negative impact on carbon nutrient function, likely because acidity suppresses microbial growth and activity, slows organic matter decomposition, and thereby decreases soil carbon availability. In addition, higher aluminum and iron activity in acidic soils can damage plant roots, hindering growth and reducing grassland productivity [53].

The metabolic footprint of soil omnivorous/predatory nematodes has a significant positive effect on the multifunctionality of grassland ecosystems. This may be because omnivorous/predatory nematodes, as higher-order trophic organisms, promote nutrient turnover efficiency by preying on bacteria, fungi, and herbivorous nematodes, thereby enhancing the cycle of elements such as carbon, nitrogen, and phosphorus [54]. The metabolic footprint of plant-parasitic nematodes has a significant negative effect on carbon nutrient function. This may be because plant-parasitic nematodes destroy the absorption and transportation function of the root system, resulting in a decrease in the distribution of plant photosynthetic products to the underground, thereby reducing the input of soil organic carbon [55]. The metabolic footprint of fungivorous nematodes has a significant positive effect on nitrogen nutrient function. This may be because the feeding behavior of nematodes stimulates the metabolic activity of fungi, causing fungi to secrete more nitrogen-containing metabolites, which are then absorbed and utilized by plants [56].

## 5. Conclusions

This study demonstrates that plateau zokor mound activity, by creating heterogeneous soil microhabitats, significantly drives the optimization of soil nematode community structure and function. The increase in nematode abundance and metabolic footprints under PM treatment was attributed to improved gas diffusion and microbial proliferation facilitated by the loosened mound structure, as well as rhizosphere edge effects supporting omnivorous/predatory nematodes. In contrast, the undisturbed CK plots maintained the highest carbon, nitrogen, and phosphorus functions due to organic matter accumulation and microbial stability, whereas LM plots exhibited the lowest multifunctionality as a result of hindered biotic turnover. Notably, soil moisture and pH emerged as key regulators of nitrogen–phosphorus cycling and carbon transformation, respectively, underscoring the fundamental role of hydrochemical coupling in ecosystem functioning. These findings reveal the mechanisms by which *zokor mounds* with different vegetation types regulate ecosystem multifunctionality through nematode community structures. This provides practical guidance for the restoration of degraded alpine meadows, ultimately contributing to improved soil health and the sustainable management of grassland ecosystems.

## Figures and Tables

**Figure 1 biology-14-01200-f001:**
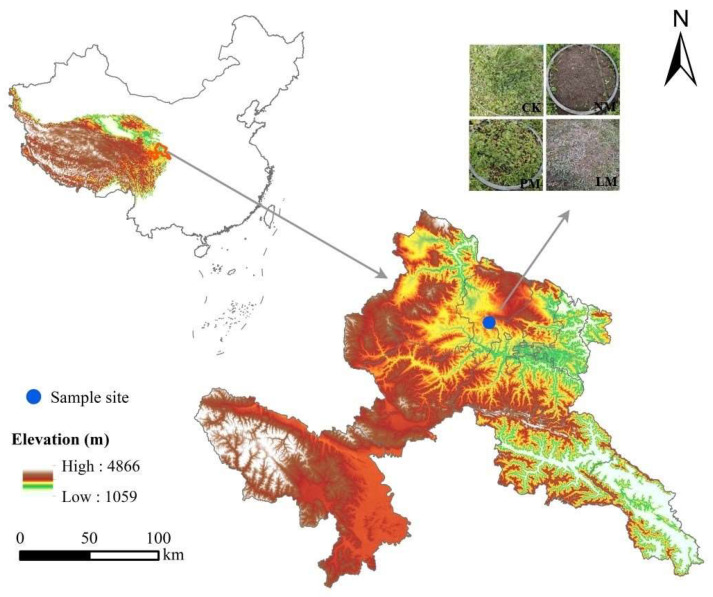
Location of the study site.

**Figure 2 biology-14-01200-f002:**
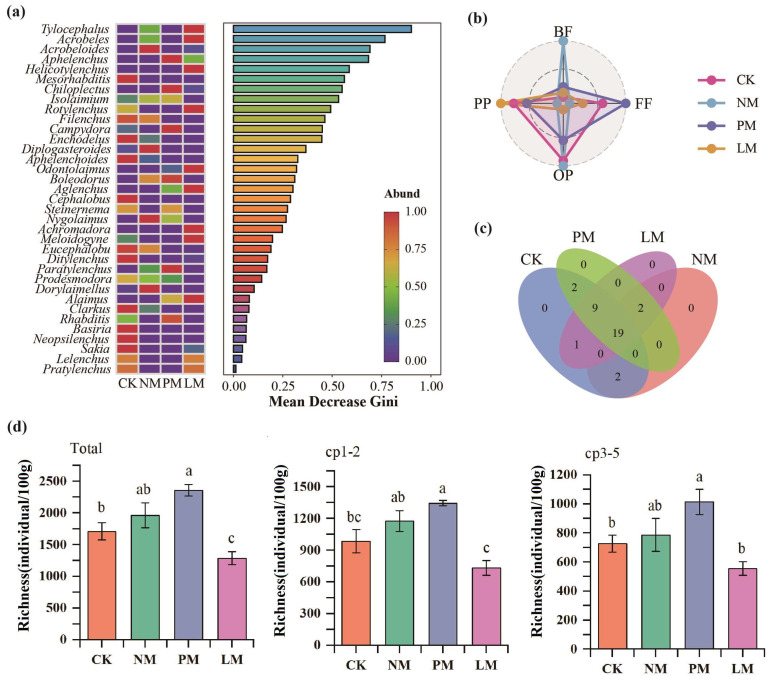
Analysis of soil nematode community composition and trophic groups. (**a**) random forest analysis; (**b**) relative abundance analysis of trophic groups; (**c**) Venn analysis. (**d**) Richness and abundance of soil nematodes. BF—bacterivorous nematodes; FF—fungivorous nematodes; PP—plant-parasitic nematodes; OP—omnivorous/predatory nematodes. CK—Control treatment; NM—Bare new *zokor mounds*; PM—Potentilla anserina *zokor mounds*; LM—Leontopodium leontopodioides *zokor mounds*. Different letters indicate significant differences among treatments (*p* < 0.05).

**Figure 3 biology-14-01200-f003:**
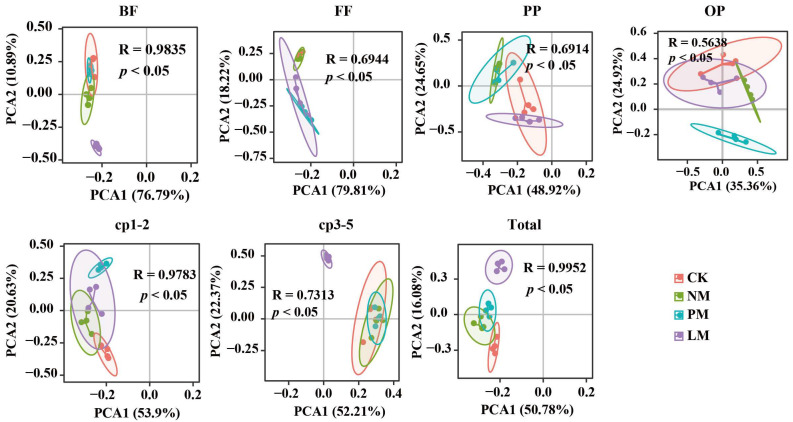
PCA analysis of soil nematode trophic groups, CP groups, and total nematode communities. Note: *p* and R values are the results of ANOSIM test, *p* < 0.05 indicates significant difference, R > 0 indicates difference. BF—bacterivorous nematodes; FF—fungivorous nematodes; PP—plant-parasitic nematodes; OP—omnivorous/predatory nematodes. CK—Control treatment; NM—Bare new *zokor mounds*; PM—Potentilla anserina *zokor mounds*; LM—Leontopodium leontopodioides *zokor mounds*.

**Figure 4 biology-14-01200-f004:**
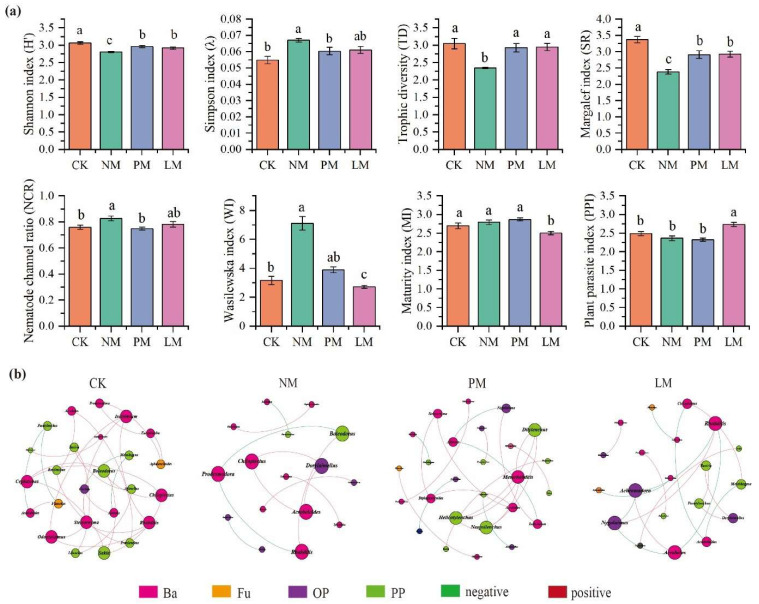
Diversity indices and co-occurrence network of soil nematode communities. (**a**) Diversity indices of nematode communities, with different lowercase letters denoting significant differences among treatments (*p* < 0.05). (**b**) Co-occurrence network of nematode communities, where node size reflects degree (number of connections), red and green edges indicate positive and negative correlations, respectively, and node colors represent distinct taxonomic groups. CK—Control treatment; NM—Bare new *zokor mounds*; PM—Potentilla anserina *zokor mounds*; LM—Leontopodium leontopodioides *zokor mounds*.

**Figure 5 biology-14-01200-f005:**
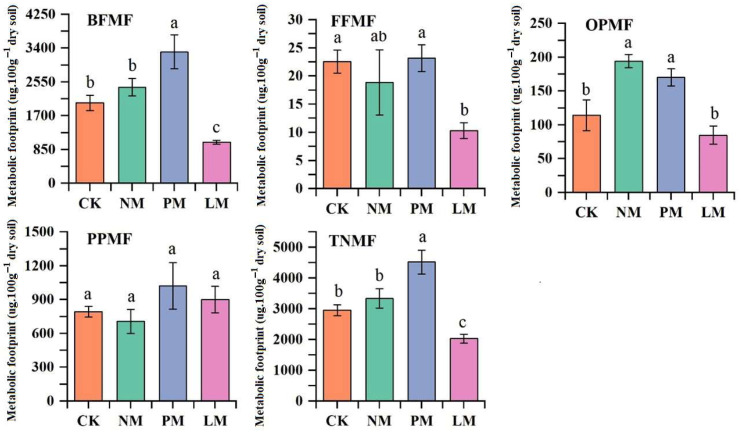
Comparison of metabolic footprints of different trophic groups of soil nematodes. Note: Different lowercase letters indicate significant differences among treatments (*p* < 0.05). BFMF: metabolic footprint of bacterivorous nematodes; FFMF: metabolic footprint of fungivorous nematodes; PPMF: metabolic footprint of plant-parasitic nematodes; OPMF: metabolic footprint of omnivorous/predatory nematodes; TNMF: total metabolic footprint of nematodes. CK—Control treatment; NM—Bare new *zokor mounds*; PM—Potentilla anserina *zokor mounds*; LM—Leontopodium leontopodioides *zokor mounds*.

**Figure 6 biology-14-01200-f006:**
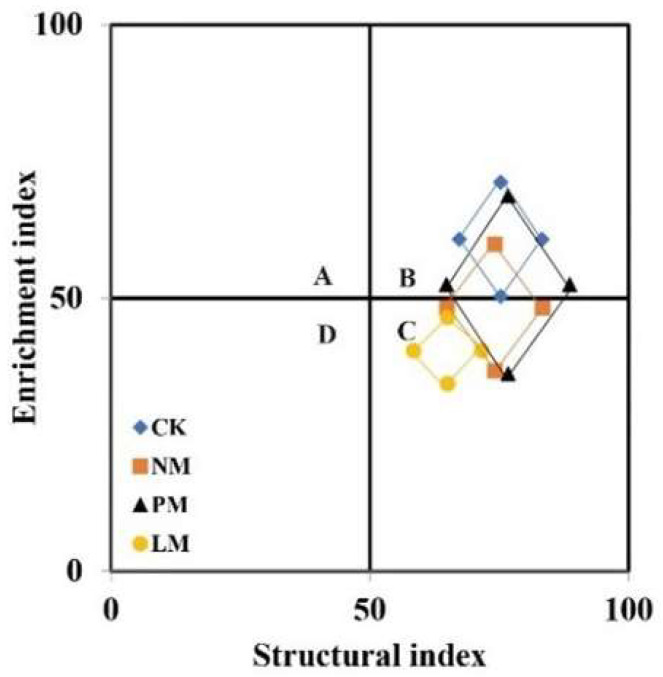
Analysis of soil nematode community flora. CK—Control treatment; NM—Bare new *zokor mounds*; PM—Potentilla anserina *zokor mounds*; LM—Leontopodium leontopodioides *zokor mounds*.

**Figure 7 biology-14-01200-f007:**
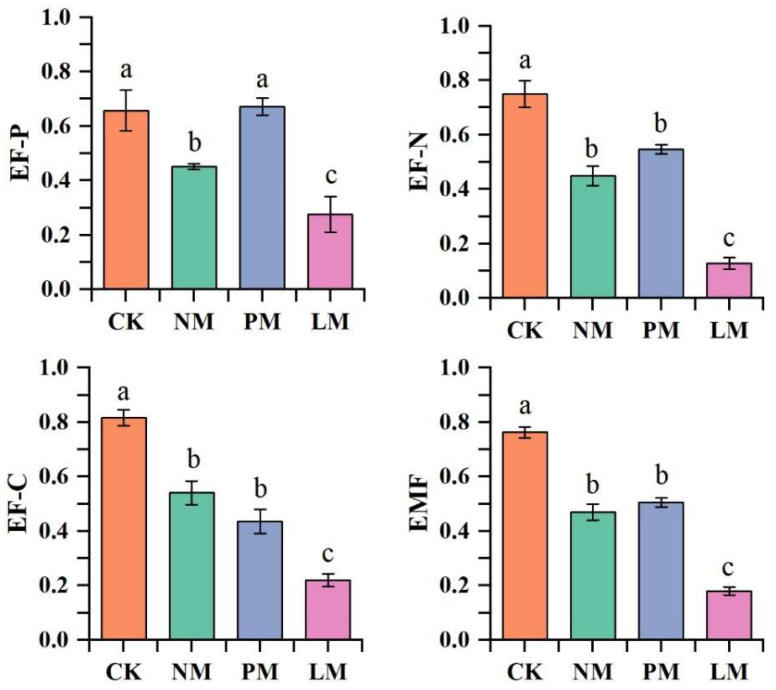
Soil ecosystem multifunctionality. Note: EF-C: soil carbon nutrient; EF-N: soil nitrogen nutrient; EF-P: soil phosphorus nutrient; EMF: ecosystem multifunctionality index. Note: Different lowercase letters indicate the difference level of soil bacteria and soil fungi diversity index between different nitrogen fertilizer addition treatments (*p* < 0.05). Same below. CK—Control treatment; NM—Bare new *zokor mounds*; PM—Potentilla anserina *zokor mounds*; LM—Leontopodium leontopodioides *zokor mounds*.

**Figure 8 biology-14-01200-f008:**
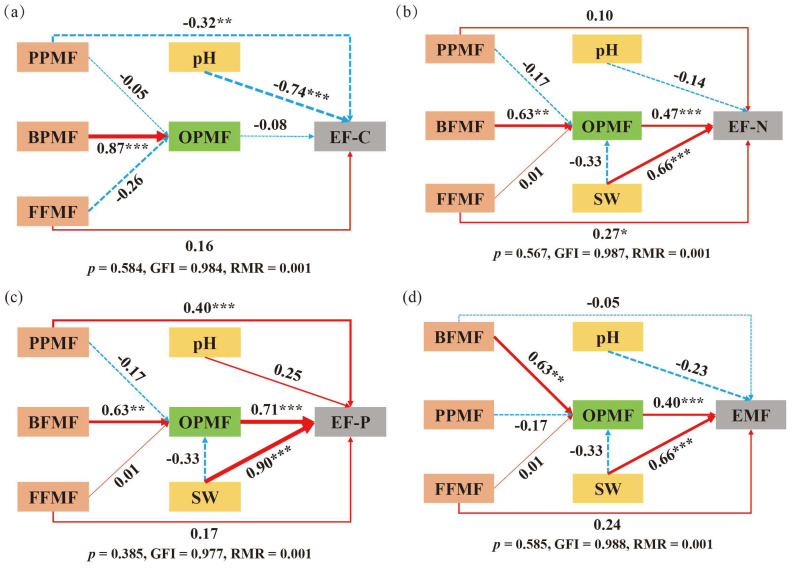
Structural equation of soil nutrient function. (**a**) represents the EF-C structural equation; (**b**) represents the EF-N structural equation; (**c**) represents the EF-P structural equation; (**d**) represents the EMF structural equation. Note: Red solid arrows represent significant positive correlations, while blue dashed arrows denote negative correlations. Numbers on the arrows indicate standardized path coefficients. Significance levels: * *p* < 0.05, ** 0.001 < *p* < 0.01, and *** *p* < 0.001. SW: soil water content; EF-C: soil carbon nutrient; EF-N: soil nitrogen nutrient; EF-P: soil phosphorus nutrient; EMF: ecosystem multifunctionality index. BFMF: metabolic footprint of bacterivorous nematodes; FFMF: metabolic footprint of fungivorous nematodes; PPMF: metabolic footprint of plant-parasitic nematodes; OPMF: metabolic footprint of omnivorous/predatory nematodes.

**Figure 9 biology-14-01200-f009:**
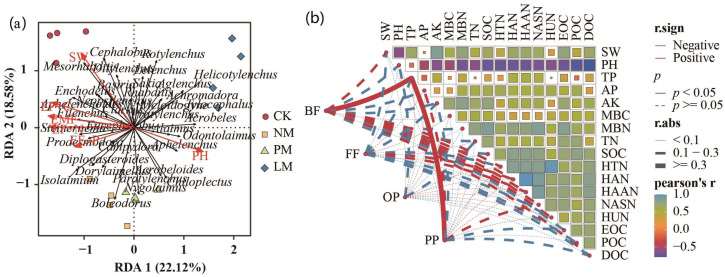
The relationship between soil nematodes and soil functions. (**a**) represents the redundancy analysis; (**b**) represents the Mantel analysis.

## Data Availability

The data presented in this study are available on request from the corresponding author. The data are not publicly available due to privacy.

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
