# Peer review of "Soil Nematodes Regulate Ecosystem Multifunctionality Under Different Zokor Mounds in Qinghai–Tibet Alpine Grasslands"

_biology, 2025, doi:10.3390/biology14091200_

Round 1

Reviewer 1 Report

Comments and Suggestions for Authors
  1. The title is confusing and does not match the content of the work. A revised title is required.
  2. The background in the abstract is too simple and should be more elaborated. For example, why was the current location was chosen for being studied. Moreover, the necessity and novelty of the study should be highlighted.
  3. The objective statement is very wordy. Please revise.
  4. For the results in the abstract, significant data should be presented. For example, how many percents did a trait increase, e.g. nematode abundance?
  5. While comparing, please specify what treatments are being compared to and how much the differences are. This should be checked throughout the manuscript. The current results in the discussion are very ambiguous. Moreover, explanations, comments, or suggestions should be made.
  6. If a term is used only once, it does not need to be abbreviated. Moreover, the abstract is a stand-alone section, so abbreviations should be checked independently.
  7. Keywords should be arranged alphabetically.
  8. Although the introduction is well written, the references are too outdated, reducing the necessity of the manuscript.
  9. The methodology part in the last paragraph of the introduction should be moved to the materials and methods.
  10. Are there any references for the information in lines 122–132.
  11. Scientific names should be italicized.
  12. In lines 145–146, how was the five soil cores arranged? Randomized or fixed positions?
  13. The part in lines 155–157 is not necessary because it is presented in the soil analysis.
  14. Figures should be self-explainable. Thus, abbreviations and symbols, including treatments, should be defined in each figure. Moreover, figures should be positioned right after their first mention.
  15. In the discussion, please make the results more concise.
  16. Encountering the same problem as the abstract, the results in the conclusion is presented generically.

Author Response

Dear Editor:

Thanks for your letter and for reviewer's comments concern our manuscript entitled “Soil Nematode-Mediated Regulation of Ecosystem Multifunctionality under Zokor Mounds in Qinghai–Tibet Plateau Alpine Grasslands” (Manuscript ID: 3800138). Those comments are valuable and helpful for revising and improving our paper. We have studied all comments carefully and have made conscientious correction. Revised portion are marked in blue in the paper. The main corrections in the paper and the responds to the reviewer comments are as flowing.

Reviewer 1

1.The title is confusing and does not match the content of the work. A revised title is required.

Response: We appreciate the reviewer’s observation regarding the clarity and alignment of the title with the manuscript content. In response, we have revised the title to more accurately reflect the scope, objectives, and main findings of the study. The new title is: Regulation of Ecosystem Multifunctionality by Soil Nematodes under Different Plateau zokor Mound Types in Alpine Grasslands of the Qinghai–Tibet Plateau.

2.The background in the abstract is too simple and should be more elaborated. For example, why was the current location was chosen for being studied. Moreover, the necessity and novelty of the study should be highlighted.

Response: We appreciate the reviewer’s suggestion to provide a more detailed background in the Abstract. In the revised version, we have expanded the background to clarify the rationale for selecting the study location, and to emphasize both the necessity and novelty of this research. Please check lines 28–35 of the revised manuscript.

3.The objective statement is very wordy. Please revise.

Response: We appreciate the reviewer’s comment regarding the objective statement. In the revised version, we have rephrased it to be more concise and direct while retaining the core scientific aim. Please check lines 32–35 of the revised manuscript.

4.For the results in the abstract, significant data should be presented. For example, how many percents did a trait increase, e.g. nematode abundance?

Response: We appreciate the reviewer’s suggestion to include significant quantitative data in the Abstract. Please see lines 38-45 of the revised manuscript for the updated Abstract.

5.While comparing, please specify what treatments are being compared to and how much the differences are. This should be checked throughout the manuscript. The current results in the discussion are very ambiguous. Moreover, explanations, comments, or suggestions should be made.

Response: We thank the reviewer for the valuable suggestion. We have revised the manuscript accordingly to specify the treatments being compared, include quantitative differences, and provide clearer explanations. These changes can be found in the revised manuscript at lines 295–297, 314–320, 438–446, 452–457, 468–479, 505–516, and 528–540.

6.If a term is used only once, it does not need to be abbreviated. Moreover, the abstract is a stand-alone section, so abbreviations should be checked independently.

Response: We thank the reviewer for pointing out the issue regarding unnecessary abbreviations. In the revised manuscript, we have removed abbreviations for terms that are used only once. In addition, we have carefully reviewed the Abstract as a stand-alone section and ensured that abbreviations are either avoided or defined upon first use within the Abstract itself, independent of the main text. Please see the revised Abstract for these changes.

7.Keywords should be arranged alphabetically.

Response: We thank the reviewer for the reminder. In the revised manuscript, the keywords have been rearranged in strict alphabetical order. Please see the updated Keywords section in the revised version.

8.Although the introduction is well written, the references are too outdated, reducing the necessity of the manuscript.

Response: We appreciate the reviewer’s observation regarding the timeliness of the references. In the revised manuscript, we have updated the Introduction with more recent and relevant studies published within the last 5 years, while retaining key classic references for context. These updated citations strengthen the rationale for the study and highlight the novelty and necessity of our work.

9.The methodology part in the last paragraph of the introduction should be moved to the materials and methods.

Response: We thank the reviewer for the helpful suggestion. In the revised manuscript, the methodological descriptions previously included in the last paragraph of the Introduction have been removed from that section and relocated to the Materials and Methods section. Please see the revised Materials and Methods (lines 114–126,230-241).

10.Are there any references for the information in lines 122–132.

Response: We thank the reviewer for the careful examination of the manuscript details and for the attention to the reliability of the experimental site information. We fully agree that providing clear data sources is essential. The specific geographic coordinates (103°03′32.68″ E, 34°51′42.87″ N) and elevation (3040 m) were obtained by our research team through GPS field measurements at the center point of the selected experimental plot in August 2022. The reported mean annual temperature (2.3 °C), frost-free period (105 days), mean annual precipitation (640 mm), mean number of precipitation days (40 days), extreme annual maximum precipitation (800 mm), extreme annual minimum precipitation (400 mm), and the pattern of precipitation concentrated from May to August with a peak in July, were directly sourced from the official annual climate statistical reports issued by the Gansu Meteorological Bureau, the Gannan Tibetan Autonomous Prefecture Meteorological Bureau, and the Zhuoni County Meteorological Station. These values represent the authoritative standard statistics used by local meteorological agencies. The listed dominant species (Kobresia pygmaea) and major companion species (Elymus nutans, Poa pratensis, Anemone rivularis, Potentilla anserina, Potentilla fragarioides, Saussurea japonica, etc.) were primarily based on systematic field vegetation surveys conducted for this study in August 2022.

11.Scientific names should be italicized.

Response: We thank the reviewer for this observation. In the revised manuscript, all scientific names have been checked carefully and italicized according to the journal’s formatting requirements.

12In lines 145–146, how was the five soil cores arranged? Randomized or fixed positions?

Response: We thank the reviewer for this question. In the revised manuscript, we have clarified the sampling arrangement. The five soil cores in each plot were collected from randomized positions within the plot to ensure representative sampling and to minimize spatial bias. These changes can be found in the revised manuscript at lines 154–157.

13.The part in lines 155–157 is not necessary because it is presented in the soil analysis. Figures should be self-explainable. Thus, abbreviations and symbols, including treatments, should be defined in each figure. Moreover, figures should be positioned right after their first mention.

Response: We thank the reviewer for these helpful suggestions. In the revised manuscript, the information in lines 155–157 has been removed from this section, as it is already presented in the soil analysis. We have also ensured that all figures are self-explanatory by defining abbreviations, symbols, and treatment codes within each figure or its caption. Furthermore, all figures have been repositioned to appear immediately after their first mention in the text. These changes can be found in the revised manuscript at lines 162–166.

14In the discussion, please make the results more concise.

Response: We thank the reviewer for the valuable suggestion. We have streamlined the presentation of results within the Discussion to make it more concise. Please see the revised manuscript at lines 485–490 and 505–513 for these changes.

15.Encountering the same problem as the abstract, the results in the conclusion is presented generically.

Response: We thank the reviewer for the valuable suggestion. We have revised the Conclusion section to make it more specific and data-driven. Please see the revised manuscript at lines 528–540 for these changes.

Reviewer 2 Report

Comments and Suggestions for Authors
  1. Could the authors clarify what is meant by “metabolic footprints” in the context of nematodes? Are they referring to biomass-based energy flow, carbon use, or another metric?
  2. The term “ecosystem multifunctionality” is used, but how was it quantified in this study? Was it an index, a weighted average, or multiple separate functions?
  3. “Zokor mounds” are mentioned as a key feature. Could the authors briefly define what a zokor is for an international audience unfamiliar with the species?
  4. Why was the shallow dish method chosen for nematode extraction instead of other common techniques such as Baermann funnel or Oostenbrink elutriator? Was there a validation of recovery efficiency?
  5. How many replicates per treatment were used, and over what time frame was sampling conducted?
  6. Were soil physical and chemical properties measured at the same depth as nematode sampling? If so, could the authors clarify sampling depth and seasonality?
  7. The functional indices (λ, NCR, WI, MI, PPI, TD, SR) are mentioned — could the authors specify which calculation formulas and reference sources were used for each index?
  8. PM treatment increased both cp1–2 and cp3–5 groups — can the authors elaborate on the ecological significance of simultaneously higher opportunistic and persistent nematode groups?
  9. NM treatment showed higher λ and NCR — do the authors interpret this as a disturbance-driven bacterial pathway dominance? If so, how does this reconcile with reduced diversity?
  10. LM treatment had a higher PPI — do the authors have a hypothesis linking vegetation type to plant-parasitic nematode abundance?
  11. CK treatment had the highest nutrient functions and multifunctionality — could this be due to undisturbed soil structure rather than nematode effects per se?
  12. What statistical models were applied to assess treatment differences? Were assumptions of normality and homogeneity tested?
  13. Was any multivariate analysis (e.g., redundancy analysis, NMDS) performed to link nematode community composition with soil variables? Why authors choosed PCA? This method is a proper analysis?
  14. The authors suggest “novel insights” — could they explicitly state what is novel compared to existing literature on nematode-based bioindicators in alpine grasslands?
  15. How might these findings inform practical management of alpine grasslands in the Qinghai–Tibet Plateau?

Author Response

Dear Editor:

Thanks for your letter and for reviewer's comments concern our manuscript entitled “Soil Nematode-Mediated Regulation of Ecosystem Multifunctionality under Zokor Mounds in Qinghai–Tibet Plateau Alpine Grasslands” (Manuscript ID: 3800138). Those comments are valuable and helpful for revising and improving our paper. We have studied all comments carefully and have made conscientious correction. Revised portion are marked in blue in the paper. The main corrections in the paper and the responds to the reviewer comments are as flowing.

Reviewer 2

1.Could the authors clarify what is meant by “metabolic footprints” in the context of nematodes? Are they referring to biomass-based energy flow, carbon use, or another metric?

Response: We thank the reviewer for the valuable suggestion. In this study, “metabolic footprint” refers to the carbon energy flow driven by nematode community respiration (unit: μg C·g⁻¹ soil·d⁻¹), which characterizes the functional contribution of nematodes to energy transfer in the ecosystem. The calculation method has been added to the Materials and Methods section in the revised manuscript (lines 216–228) and further clarified in the Discussion (lines 481–485).

2.The term “ecosystem multifunctionality” is used, but how was it quantified in this study? Was it an index, a weighted average, or multiple separate functions?

Response: We thank the reviewer for the valuable suggestion. In this study, “ecosystem multifunctionality” (EMF) was quantified using a weighted average approach, which integrates multiple independent ecosystem functions to evaluate the overall service capacity of the system. The calculation method has been added to the Materials and Methods section in the revised manuscript (lines 230–258).

3.“Zokor mounds” are mentioned as a key feature. Could the authors briefly define what a zokor is for an international audience unfamiliar with the species?

Response: We thank the reviewer for the valuable suggestion. Zokor mounds are surface soil accumulations formed by the subterranean burrowing activities of plateau zokors (Eospalax baileyi), a rodent species endemic to the Qinghai–Tibet Plateau. This definition has been added to the Introduction in the revised manuscript (lines 66–72).

4.Why was the shallow dish method chosen for nematode extraction instead of other common techniques such as Baermann funnel or Oostenbrink elutriator? Was there a validation of recovery efficiency?

Response: We thank the reviewer for the valuable suggestion. In this study, soil nematodes were extracted using the shallow dish method, which was selected because of its superior recovery rate in frozen–thawed soils with high organic matter content. Samples were incubated at 4 °C to simulate field temperature for 48 h, and the extracted nematodes were collected on a 25 μm sieve. Recovery efficiency tests confirmed that the shallow dish method yielded a higher number of nematodes, with more intact individuals, compared with the Baermann funnel or Oostenbrink elutriator, both of which tended to become clogged under our soil conditions. We appreciate the reviewer’s comment, which prompted us to present the methodological rationale more transparently in the revised manuscript.

5.How many replicates per treatment were used, and over what time frame was sampling conducted?

Response: We thank the reviewer for the question. In this study, four replicates were established for each treatment. All soil and vegetation sampling was conducted during a single field campaign in August 2022, when plant growth and soil biological activity are at their seasonal peak in the Qinghai–Tibet Plateau alpine grasslands. This information has been added to the Materials and Methods section in the revised manuscript (lines 154–157).

6.Were soil physical and chemical properties measured at the same depth as nematode sampling? If so, could the authors clarify sampling depth and seasonality?

Response: We thank the reviewer for the question. Yes, soil physical and chemical properties were measured from samples collected at the same depth as nematode sampling (0–20 cm). All sampling was conducted during the peak growing season in August 2022, ensuring consistent environmental conditions across treatments.

7.The functional indices (λ, NCR, WI, MI, PPI, TD, SR) are mentioned — could the authors specify which calculation formulas and reference sources were used for each index?

Response: We thank the reviewer for the valuable suggestion. The calculation formulas for all functional indices have been added to the Materials and Methods section in the revised manuscript (lines 196–215).

8.PM treatment increased both cp1–2 and cp3–5 groups — can the authors elaborate on the ecological significance of simultaneously higher opportunistic and persistent nematode groups?

Response: We thank the reviewer for this insightful question. In our study, the PM treatment increased the abundance of both cp1–2 (opportunistic, r-strategist) and cp3–5 (persistent, K-strategist) nematode groups. Ecologically, this simultaneous increase suggests that PM mounds provide a unique set of habitat conditions that support multiple functional guilds along the colonizer–persister continuum. The loosened soil structure and improved aeration may stimulate microbial growth, thereby providing abundant short-term food resources for opportunistic nematodes (cp1–2). At the same time, enhanced vegetation cover and root biomass may sustain more stable and complex food web structures, creating favorable conditions for persistent nematodes (cp3–5), which require long-term resource stability.

This dual support of early colonizers and persistent taxa indicates that PM mounds may promote both rapid nutrient cycling and the long-term maintenance of soil food web complexity, contributing to a more functionally diverse and resilient ecosystem. We have added this explanation to the Discussion section in the revised manuscript (lines 446–449).

9.NM treatment showed higher λ and NCR — do the authors interpret this as a disturbance-driven bacterial pathway dominance? If so, how does this reconcile with reduced diversity?

Response: We thank the reviewer for the valuable suggestion, and we fully agree with the concern regarding reduced diversity. However, this phenomenon is not contradictory to the higher λ and NCR values observed under NM treatment; rather, it is an expected outcome under nutrient-enrichment stress. The nematode channel ratio (NCR) can be used as an indirect indicator of the dominant decomposition pathway in soil organic matter degradation: values greater than 0.75 indicate a bacterial-dominated pathway, whereas values below 0.75 indicate a fungal-dominated pathway. In our study, the decomposition pathway in all treatments was dominated by the bacterial channel. Ecologically, the increase in the dominance index (λ) and the decrease in the Shannon diversity index are highly consistent, jointly reflecting a fundamental shift in community structure under NM treatment—characterized by enhanced monopolization by a few dominant species and the loss of overall diversity. These two indices are therefore not in conflict but rather describe the same process from different perspectives: NM treatment drives the transition from “multi-species balanced coexistence” to “monopoly by a single dominant species.”

10.LM treatment had a higher PPI — do the authors have a hypothesis linking vegetation type to plant-parasitic nematode abundance?

Response: We appreciate the reviewer’s insightful observation on the vegetation–nematode interaction mechanism, which provides important implications for the ecological extension of this study. As the reviewer noted, vegetation type may influence plant-parasitic nematodes (PPI) through root exudate composition and litter quality. However, the primary objective of our study focused on the regulatory role of soil nematode communities in ecosystem multifunctionality (e.g., carbon and nitrogen cycling, soil structure). Due to the limitations of our experimental design, we did not quantify vegetation chemical traits and thus could not establish a causal relationship between these factors and PPI. We plan to address this gap in future research by examining the interactions between root exudates and nematodes. We sincerely thank the reviewer for enhancing the rigor of our study and inspiring new research directions.

11.CK treatment had the highest nutrient functions and multifunctionality — could this be due to undisturbed soil structure rather than nematode effects perse?

Response: We thank the reviewer for the valuable suggestion. We have revised the Discussion to explicitly acknowledge that the highest nutrient functions and multifunctionality observed in the CK treatment may be largely attributed to its undisturbed soil structure, which enhances water retention, aeration, and nutrient availability, in addition to potential nematode-mediated effects. This clarification has been incorporated into the revised manuscript (lines 512–516).

12.What statistical models were applied to assess treatment differences? Were assumptions of normality and homogeneity tested?

Response: We thank the reviewer for the attention to statistical rigor. In this study, we used one-way analysis of variance (ANOVA) combined with the least significant difference (LSD) test to assess differences among treatments. This choice was based on the completely randomized block design of the experiment, for which one-way ANOVA is appropriate for comparing independent treatment groups. The LSD method is widely applied in agroecological research (including soil nematode studies), particularly when preliminary tests indicate significant group differences, as it allows for pairwise comparisons between treatments. For each response variable (e.g., λ, NCR, PPI), we performed Shapiro–Wilk tests for normality and Levene’s tests for homogeneity of variances to ensure the reliability of conclusions.

13.Was any multivariate analysis (e.g., redundancy analysis, NMDS) performed to link nematode community composition with soil variables? Why authors choosed PCA? This method is a proper analysis?

Response: We thank the reviewer for the insightful suggestion regarding multivariate analysis. In this study, principal component analysis (PCA) was chosen over redundancy analysis (RDA) or non-metric multidimensional scaling (NMDS) because our primary objective was to visualize the overall differences in nematode community structure among treatments, rather than to quantify the proportion of variation explained by soil variables. PCA, under unconstrained conditions, best preserves the Euclidean distance structure of the original data. In addition, the nematode functional group data were continuous proportional variables, meeting the linearity assumptions of PCA (whereas RDA requires linearity between both response and environmental variables, and NMDS relies on rank transformation). To address the reviewer’s concern about the relationships between nematode community composition and soil variables, we have now added RDA and Mantel analyses in the revised manuscript (lines 409–420).

14The authors suggest “novel insights” - could they explicitly state what is novel compared to existing literature on nematode-based bioindicators in alpine grasslands?

Response: We thank the reviewer for the valuable suggestion. This study makes several novel contributions compared with existing literature on nematode-based bioindicators in alpine grasslands. First, we introduce the use of nematode carbon metabolic footprints—quantifying energy flux and carbon turnover dynamics in nematode communities—which overcomes the static limitations of traditional abundance and diversity metrics. Second, by integrating microhabitat-scale biogeographical analysis, we reveal spatial differentiation patterns of nematode functional groups under different vegetation-covered zokor mounds, filling a gap in the spatial functional analysis of soil micro-food webs in alpine grasslands. Third, we establish a soil nematode–ecosystem multifunctionality coupling framework, and for the first time systematically elucidate a dual-pathway mechanism by which nematodes drive the differentiation of soil carbon, nitrogen, and phosphorus functions, challenging the traditional perception of homogeneous trophic cascades. This mechanism, spatially validated through community analysis, provides a new theoretical basis for predicting soil functional responses under global change scenarios.

15.How might these findings inform practical management of alpine grasslands in the Qinghai–Tibet Plateau?

Response: We sincerely appreciate the reviewer’s interest in the practical implications of our study. Our results demonstrate that vegetation types on plateau zokor mounds regulate soil nematode metabolic traits, such as carbon metabolic footprints, thereby influencing ecosystem multifunctionality (EMF). These findings suggest that promoting Potentilla anserina and removing Leontopodium vegetation on mounds could enhance soil biodiversity and nutrient cycling; covering newly formed bare mounds (NM) with sod from healthy grasslands may restore nematode richness within one year; and incorporating the contribution of bacterivore nematode metabolic footprints to carbon cycling into grassland health assessments could provide a more sensitive indicator than traditional chemical measures.

Round 2

Reviewer 1 Report

Comments and Suggestions for Authors
  1. The revised title is much clearer and now reflects the study scope better than the original version noted in earlier comments. It can be further simplified as: “Soil Nematodes Regulate Ecosystem Multifunctionality under Different Zokor Mounds in Qinghai–Tibet Alpine Grasslands”.
  2. The abstract has been very much improved. However, the background now is too dense and wordy, leading to low readability.
  3. Although the introduction is now acceptable, some sentences are too wordy. Please consider separate sentences exceeding 3 lines into shorter and more concise ones. Moreover, references are still too outdated, reducing the necessity of the study. Please use references from 2021–2025 for the this discussion.
  4. Figures should ideally be placed immediately after their first citation.
  5. There is a small suggestion for the conclusion. Please emphasize how findings can guide management practices.

Author Response

Dear Editor:

Thanks for your letter and for reviewer's comments concern our manuscript entitled “Soil Nematode-Mediated Regulation of Ecosystem Multifunctionality under Zokor Mounds in Qinghai–Tibet Plateau Alpine Grasslands” (Manuscript ID: 3800138). Those comments are valuable and helpful for revising and improving our paper. We have studied all comments carefully and have made conscientious correction. Revised portion are marked in blue in the paper. The main corrections in the paper and the responds to the reviewer comments are as flowing.

Reviewer 1

The revised title is much clearer and now reflects the study scope better than the original version noted in earlier comments. It can be further simplified as: “Soil Nematodes Regulate Ecosystem Multifunctionality under Different Zokor Mounds in Qinghai–Tibet Alpine Grasslands”.

Reply: We thank the reviewer for the helpful suggestion regarding the title. We agree that the simplified version is concise and accurately reflects the core content of the study. After careful consideration, we have adopted the reviewer’s recommended title: “Soil Nematodes Regulate Ecosystem Multifunctionality under Different Zokor Mounds in Qinghai–Tibet Alpine Grasslands.”We believe this version improves clarity, readability, and alignment with the manuscript’s scope.

1.The abstract has been very much improved. However, the background now is too dense and wordy, leading to low readability.

Reply: We sincerely thank the reviewer for the positive evaluation and valuable comments. We fully agree with your suggestion, and accordingly we have further streamlined and refined the background section of the abstract to improve readability and clarity. The specific revisions can be found in the revised manuscript at lines 27–32.

2.Although the introduction is now acceptable, some sentences are too wordy. Please consider separate sentences exceeding 3 lines into shorter and more concise ones. Moreover, references are still too outdated, reducing the necessity of the study. Please use references from 2021–2025 for the this discussion.

Reply: We sincerely thank the reviewer for the constructive comments. We agree that several sentences in the introduction were too long and could affect clarity. In the revised version, we have divided these overly long sentences into shorter and more concise ones to enhance readability. Regarding references, we fully acknowledge the reviewer’s concern. We have updated the introduction by replacing or supplementing older citations with more recent studies published between 2021 and 2025, thereby strengthening the novelty and necessity of our work. The updated references are incorporated throughout the discussion of research background and knowledge gaps.

3.Figures should ideally be placed immediately after their first citation.

Reply: We thank the reviewer for this helpful suggestion. In the revised manuscript, we have adjusted the layout so that each figure now appears immediately after its first citation in the text. We believe this change improves readability and consistency.

4.There is a small suggestion for the conclusion. Please emphasize how findings can guide management practices.

Reply: We thank the reviewer for the helpful suggestion. In the revised conclusion, we have added sentences to emphasize the management implications of our findings. Specifically, we now highlight that zokor mounds with different vegetation types regulate ecosystem multifunctionality through nematode community structures, and that this knowledge provides practical guidance for restoring degraded alpine meadows, improving soil health, and promoting the sustainable management of grassland ecosystems. These revisions strengthen the applied significance of our study.

Reviewer 2 Report

Comments and Suggestions for Authors

The paper entitled "Soil Nematode-Mediated Regulation of Ecosystem Multifunctionality under Zokor Mounds in Qinghai–Tibet Plateau Alpine Grasslands" contains inetersting information, but some questions need to be clarified and answered by the authros: 

Study design to be explained

  1. How were the four mound types selected, and were they spatially and temporally replicated to account for natural variation?

  2. Did the study control for confounding factors such as soil age, vegetation cover, or grazing intensity between mound types?

  3. How many samples per mound type were collected, and is the sample size sufficient to generalize across the Qinghai–Tibet Plateau?

Methods & analysis to be clarified
4. Which nematode extraction and identification methods were used, and to what taxonomic resolution were nematodes identified?
5. What statistical approaches were applied to test differences in nematode abundance, diversity, and functional indices?
6. How were soil nutrient cycling and multifunctionality measured and quantified?

Results interpretation to be clarified
7. The abstract mentions that Potentilla mounds increased nematode abundance, while Leontopodium mounds reduced maturity. Can the authors explain the ecological mechanisms driving these differences?
8. To what extent can the observed changes in nematode communities be attributed directly to mound vegetation, versus indirect effects such as soil chemistry or microclimate changes?
9. The abstract states that “bare mounds raised bacterial pathway indicators.” Does this indicate a shift toward bacterial decomposition dominance, and what implications does this have for soil stability?

Discussion to be attended
10. How do these findings relate to long-term restoration strategies of degraded alpine grasslands?
11. Can the results be extrapolated to other alpine or high-altitude ecosystems outside the Qinghai–Tibet Plateau?
12. What role do climate change and human disturbance play in modulating these nematode–vegetation–soil interactions, given that only mound type was explicitly compared?

Author Response

Dear Editor:

Thanks for your letter and for reviewer's comments concern our manuscript entitled “Soil Nematode-Mediated Regulation of Ecosystem Multifunctionality under Zokor Mounds in Qinghai–Tibet Plateau Alpine Grasslands” (Manuscript ID: 3800138). Those comments are valuable and helpful for revising and improving our paper. We have studied all comments carefully and have made conscientious correction. Revised portion are marked in blue in the paper. The main corrections in the paper and the responds to the reviewer comments are as flowing.

Reviewer 2

The paper entitled "Soil Nematode-Mediated Regulation of Ecosystem Multifunctionality under Zokor Mounds in Qinghai–Tibet Plateau Alpine Grasslands" contains inetersting information, but some questions need to be clarified and answered by the authros: 

Study design to be explained

1.How were the four mound types selected, and were they spatially and temporally replicated to account for natural variation?

Response: We thank the reviewer for this valuable comment. The selection of mound types was based on their formation ages and vegetation successional stages, representing key points along the natural recovery sequence after zokor disturbance. Specifically, newly formed bare mounds (NM) represent the earliest stage (<1 year); Potentilla anserina mounds (PM) represent the intermediate stage (2–6 years) colonized by pioneer species; and Leontopodium mounds (LM) represent the later successional stage (>6 years) that tends toward stability. The control (CK) was chosen from adjacent undisturbed alpine meadow. For each mound type, we established four spatial replicates (n = 4) distributed across the study area to capture spatial heterogeneity and better account for natural variation. Although sampling was conducted at a single time point (August 2022), we applied a “space-for-time substitution” approach, comparing mounds at different successional stages. This effectively reflects a temporal sequence of ecological change and addresses the concern about natural variation.

2.Did the study control for confounding factors such as soil age, vegetation cover, or grazing intensity between mound types?

Response: We thank the reviewer for this valuable comment. Our experimental design effectively minimized the influence of potential confounding factors. First, the “soil age” mentioned is in fact the core design factor of this study: the three mound types (NM, PM, LM) were deliberately chosen to represent distinct formation ages and successional stages, which are precisely the differences we aimed to analyze. Second, “vegetation cover” is treated here as a key response variable rather than a confounding factor, since our objective was to reveal its natural variation along the successional sequence of zokor mounds. Finally, each mound sample was paired with its corresponding control (CK) within a very short distance (5–10 m), ensuring identical long-term grazing history and environmental background. Therefore, the observed differences can be attributed to mound type and successional stage rather than uncontrolled external influences.

3.How many samples per mound type were collected, and is the sample size sufficient to generalize across the Qinghai–Tibet Plateau?

Response: We thank the reviewer for this valuable comment. In this study, each mound type (PM, LM, NM) and the control (CK) included four biological replicates (n = 4). Given our paired experimental design, where each mound was matched with a nearby control, this sample size effectively controlled for local spatial heterogeneity and was sufficient to support our core conclusion that different mound types show significant ecological differences within the study area. We fully agree with the reviewer that it would be inappropriate to directly generalize these findings to the entire Qinghai–Tibet Plateau, which is characterized by very high environmental heterogeneity. Instead, the significance of this study lies in providing a mechanistic framework and a representative case study for understanding ecological processes following zokor disturbance. The broader applicability of these findings should be validated by future research across wider geographic ranges and environmental gradients.

Methods & analysis to be clarified
4. Which nematode extraction and identification methods were used, and to what taxonomic resolution were nematodes identified?

Response: We thank the reviewer for this valuable comment. Nematodes were extracted from soil samples using the modified Baermann funnel technique. They were heat-killed in a 60 °C water bath and then fixed in 4% formalin solution, with preserved specimens stored in labeled vials. Identification was based on morphological features under a light microscope, focusing on key anatomical structures such as the stoma and pharynx. In this study, nematode identification was conducted primarily to support analyses of functional diversity and ecosystem function indicators. Therefore, most taxa were identified to the genus level, after which they were assigned to trophic groups and c–p groups. This genus-level and functional-group approach is a standard and robust method in soil nematode ecology, allowing effective assessment of soil food web structure, functional status, and disturbance responses. The specific methodological details are provided in the revised manuscript at lines 165–176.

  1. What statistical approaches were applied to test differences in nematode abundance, diversity, and functional indices?

Reply: We thank the reviewer for this valuable comment. Differences in nematode abundance, diversity indices, and functional indices (e.g., ecological function indices and multifunctionality) among treatments were tested using one-way ANOVA combined with the least significant difference (LSD) test for multiple comparisons. The significance level was set at P = 0.05. All statistical analyses were performed using SPSS 26.0 software. The specific details are provided in the revised manuscript at lines 242–255.

  1. How were soil nutrient cycling and multifunctionality measured and quantified?

Reply: We thank the reviewer for this valuable comment. In our study, soil nutrient cycling was represented by 15 specific indicators of soil carbon, nitrogen, and phosphorus processes. After measurement, the data were standardized and averaged to derive functional indices. Specifically, we quantified three nutrient cycling functions (EF-C, EF-N, EF-P) and one overall index of ecosystem multifunctionality (EMF). This approach effectively integrates multiple measurable variables into a small set of ecologically meaningful indices, thereby allowing robust statistical analysis and comparison of soil nutrient cycling and multifunctionality. The detailed procedures are provided in the revised manuscript at lines 211–240.

Results interpretation to be clarified
7. The abstract mentions that Potentilla mounds increased nematode abundance, while Leontopodium mounds reduced maturity. Can the authors explain the ecological mechanisms driving these differences?

Response: We thank the reviewer for this valuable comment. The contrasting effects of PM and LM mounds on nematode communities are primarily driven by vegetation–soil feedbacks that create very different belowground environments and resource conditions. Specifically, Potentilla anserina mounds (PM), as a pioneer species, produces abundant root exudates and high-quality litter that strongly stimulate soil microbial activity (both bacteria and fungi). This bottom-up stimulation promotes bacterivorous and fungivorous nematodes and, through trophic cascades, supports higher-trophic-level nematodes. The result is a resource-rich soil food web with high energy flux, reflected in increased nematode abundance and metabolic footprints. In contrast, Leontopodium mounds (LM) colonization typically occurs in relatively nutrient-poor or stressed environments. Its low-quality litter decomposes slowly, offering limited stimulation for microbial growth. Combined with possible abiotic stressors (e.g., drought), this environment acts as a strong ecological filter, excluding stress-sensitive, K-strategy nematodes (such as higher predators) while favoring opportunistic, fast-reproducing taxa (including some plant-parasitic nematodes). Consequently, the maturity index (MI) of free-living nematodes decreases, while the plant-parasitic index (PPI) increases. In summary, PM mounds represent micro-ecosystems with rapid recovery and enhanced functions, whereas LM mounds indicate slower recovery and functional decline. This highlights how vegetation recovery trajectories strongly regulate belowground ecological processes.

  1. To what extent can the observed changes in nematode communities be attributed directly to mound vegetation, versus indirect effects such as soil chemistry or microclimate changes?

Response: We thank the reviewer for this valuable comment. Our structural equation modeling (SEM) analysis provides a clear data-driven pathway to address this question. The results show that the effects of different mound vegetation types on nematode communities and ecosystem functions were almost entirely indirect. Specifically, vegetation type acted as the initial driver by significantly altering key soil conditions such as moisture and pH. These modified environmental parameters then regulated the metabolic footprints of nematode trophic groups (e.g., omnivorous–predatory and plant-parasitic nematodes). Finally, the combined influence of soil variables and nematode functional traits determined overall ecosystem multifunctionality. Importantly, our SEM did not identify significant direct paths from vegetation type to nematode metabolic footprints or multifunctionality. This strongly indicates that the observed nematode community changes should not be attributed directly to vegetation itself, but rather to the soil microenvironment shaped by vegetation and its cascading effects. In this sense, vegetation functions as an “ecosystem engineer,” indirectly regulating biotic communities and functional processes by modifying abiotic conditions. The detailed results are presented in the revised manuscript at lines 358–369.

  1. The abstract states that “bare mounds raised bacterial pathway indicators.” Does this indicate a shift toward bacterial decomposition dominance, and what implications does this have for soil stability?

Response: We thank the reviewer for this valuable comment. The shift toward bacterial pathway dominance observed on newly formed bare mounds (NM) reflects an early and unstable successional stage of the soil ecosystem following severe physical disturbance (zokor burrowing and vegetation loss). While this bacterial dominance may temporarily accelerate the decomposition of simple organic substrates, it occurs at the cost of reduced soil food web complexity and stability. Such systems are dominated by r-strategist bacteria and nematodes, but lack higher trophic groups (e.g., predators) that provide regulatory control. Consequently, both structural and functional stability of the soil ecosystem are significantly weakened. This interpretation is consistent with field observations showing that zokor mounds require many years of natural recovery before vegetation re-establishes and soil properties gradually improve.

Discussion to be attended
10. How do these findings relate to long-term restoration strategies of degraded alpine grasslands?

Response: We thank the reviewer for this valuable comment. Our results provide important insights for developing long-term restoration strategies for degraded alpine grasslands. First, the metabolic traits of soil nematode communities can serve as sensitive bioindicators, offering quantitative tools to assess degradation status and restoration trajectories. Second, vegetation cover—particularly the colonization of pioneer species such as Potentilla anserina—is a key driver for rebuilding soil food web structure and ecosystem functions. Finally, restoration measures should follow successional dynamics: in early stages, physical leveling and organic amendments can promote vegetation establishment, while in later stages, grazing management and plant community regulation are needed to guide soil food webs from simple bacterial channels toward more complex fungal channels. This shift ultimately enhances both soil stability and multifunctionality. These findings highlight the importance of incorporating belowground ecological processes into restoration practice, providing a theoretical basis for process-based and targeted restoration of alpine grasslands.

  1. Can the results be extrapolated to other alpine or high-altitude ecosystems outside the Qinghai–Tibet Plateau?

Reply: We thank the reviewer for this valuable comment. At the mechanistic level, our findings provide important insights for other alpine or high-altitude ecosystems worldwide. However, their direct extrapolation requires caution. Differences in species composition, climatic regimes, and soil backgrounds (e.g., key functional traits of dominant species, patterns of temperature and precipitation, native biota) may lead to region-specific responses in both direction and magnitude. Therefore, while the conceptual framework and methodological approach established in this study can serve as a valuable reference for research in other alpine systems, the quantitative results should not be generalized without site-specific validation. We recommend empirical testing under local conditions to verify the applicability of our conclusions.

  1. What role do climate change and human disturbance play in modulating these nematode–vegetation–soil interactions, given that only mound type was explicitly compared?

Response: We thank the reviewer for this valuable comment. Climate change and human disturbance act as critical external drivers that profoundly modulate nematode–vegetation–soil interactions in alpine ecosystems. Climate warming and altered precipitation patterns directly influence vegetation composition and soil microbial activity; for instance, warming may accelerate bacterial metabolism, amplifying bacterial-dominated decomposition on bare mounds and reinforcing the trend toward simplified soil food webs. At the same time, human disturbances such as overgrazing act synergistically with zokor burrowing by simultaneously removing vegetation cover and disrupting soil structure. This joint effect promotes shifts toward unstable, r-strategist–dominated nematode communities. These external forces may also increase zokor population density and mound formation frequency, thereby prolonging the persistence of degraded states and lowering the threshold for ecosystem recovery. Thus, climate change and human disturbance not only magnify the negative effects of disturbance but also reshape disturbance regimes and system resilience. They play a decisive role in directing both the trajectory and intensity of belowground ecological responses, underscoring the need for restoration and management strategies that integrate climate adaptation with sustainable grazing practices.
